# The Gut Microbiota in Kidney Transplantation: A Target for Personalized Therapy?

**DOI:** 10.3390/biology12020163

**Published:** 2023-01-20

**Authors:** Yuselys García-Martínez, Margherita Borriello, Giovanna Capolongo, Diego Ingrosso, Alessandra F. Perna

**Affiliations:** 1Department of Translational Medical Science, University of Campania “Luigi Vanvitelli”, Via Pansini, Bldg 17, 80131 Naples, Italy; 2Department of Precision Medicine, University of Campania “Luigi Vanvitelli”, Via L. De Crecchio 7, 80138 Naples, Italy

**Keywords:** kidney transplantation, gut microbiota, immunosuppressive therapy, post-transplant infection, diarrhea, allograft rejection

## Abstract

**Simple Summary:**

Kidney function is compromised by several post-transplant complications associated with immunosuppressive therapy, infections, gastrointestinal toxicity, and graft rejection in kidney transplant patients. Given the ability of the gut microbiota to influence alloimmunity, drug metabolism, infections, and gastrointestinal diseases, we studied the therapeutic potential of the gut microbiota in kidney transplantation. Specific microbial signatures have been associated with graft rejection, mycophenolate mofetil and tacrolimus metabolism, and the development of new-onset diabetes after transplantation. In addition, the abundance of gut enterobacteria has been linked to the development of urinary tract infections, while other microbial populations have a protective role in urinary and respiratory tract infections. The application of microbiota-based therapies such as fecal microbiota transplantation has successfully resolved infection and refractory diarrhea events in these patients. Current data suggest that modulating the gut microbiota could potentially contribute to personalizing immunosuppressive and post-transplant complication therapies to improve graft survival and patients’ quality of life.

**Abstract:**

Kidney transplantation improves quality of life, morbidity, and mortality of patients with kidney failure. However, integrated immunosuppressive therapy required to preserve graft function is associated with the development of post-transplant complications, including infections, altered immunosuppressive metabolism, gastrointestinal toxicity, and diarrhea. The gut microbiota has emerged as a potential therapeutic target for personalizing immunosuppressive therapy and managing post-transplant complications. This review reports current evidence on gut microbial dysbiosis in kidney transplant recipients, alterations in their gut microbiota associated with kidney transplantation outcomes, and the application of gut microbiota intervention therapies in treating post-transplant complications.

## 1. Introduction

Kidney transplantation (KT) is the treatment of choice for patients with end-stage kidney disease (ESKD). While successful KT improves patients’ quality of life, morbidity, and mortality compared to those on maintenance dialysis, integrated immunosuppressive and antimicrobial therapies increase the risk of post-transplant complications [1]. Immunosuppressants possess a narrow therapeutic index and large interpatient variability regarding dose requirements. Supratherapeutic levels can promote the development of post-transplant malignancy, nephrotoxicity, gastrointestinal toxicity, and opportunistic infections, whereas subtherapeutic levels can trigger immune rejection [2]. Clinical guidelines suggest personalizing immunosuppressive therapies according to each patient’s risk for acute rejection and adverse effects, but the accurate implementation of these therapies remains to be perfected [3].

Recently, the gut microbiota has emerged as a potential target to personalize immunosuppressive therapy and manage post-transplant complications. Gut microbial dysbiosis has been reported following KT. It is generally characterized by a loss of microbial diversity and an increase in the relative abundance of Proteobacteria compared to healthy individuals [4,5,6,7,8,9]. The impact appears to be bidirectional, as the gut microbiota is increasingly recognized to influence alloimmunity, drug metabolism, and post-transplant complications in kidney transplant recipients (KTRs) [10,11,12,13,14,15,16,17,18,19].

The balance between allograft rejection and regulation is crucial to determine long-term graft survival. The gut microbiota modulates the differentiation of regulatory T cells (Tregs), helper Th1, and Th17 cells. Furthermore, it is involved in the generation of memory alloreactive T cells and the maturation of NKT cells [20]. Gut bacteria can also metabolize immunosuppressive medications commonly used in KT, such as MMF [21]. Other studies suggest the impact of the gut microbiota on the development of post-transplant infections and diarrhea, as well as the successful treatment of these complications using microbiome-based therapies [22,23,24,25].

This review provides evidence supporting the gut microbiota as a potential therapeutic target to personalize immunosuppressive therapy and the treatment of post-transplant complications in KTRs. This paper is organized into three sections describing (i) gut microbial dysbiosis in KTRs compared to chronic kidney disease (CKD) patients and healthy individuals, (ii) alterations in the gut microbiota associated with KT outcome, and (iii) gut microbiome-based therapies to treat post-transplant complications.

## 2. Review Strategy

### 2.1. Literature Search

PubMed and Embase were searched for studies addressing the role of gut microbiota in KT outcomes up to December 2021. We used the keywords gut microbiota AND kidney transplantation, gut microbiome AND renal transplantation, gut dysbiosis AND kidney transplantation, intestinal microbiota AND kidney post-transplant complications, probiotics/prebiotics AND kidney/renal transplantation, microbiome therapy AND kidney/renal transplantation, uremic toxins and kidney transplantation, immune system AND gut microbiota”. In addition, a backward reference search on selected articles was conducted when a paper of interest was identified.

### 2.2. Eligible Criteria

Cohort studies, single-case reports, and reviews published in English studying the gut microbiota alterations in KTRs older than 18 years, post-transplant complications, immunosuppressants metabolism, uremic toxicity, and the gut microbiota as a therapeutic target were selected. Studies involving cancer patients were excluded.

### 2.3. Data Extraction

Relevant abstracts were selected for full-text review. Details regarding the study population, design, microbiome composition in KTRs, and transplantation outcome were extracted per article chosen. Out of 184 records identified, 121 abstracts were screened, 64 full texts were retrieved and assessed for eligibility, and 56 were included in the review. Of those, 23 studies reported a potential role of gut microbiota in KT. The selection process is illustrated in Figure 1.

## 3. Gut Microbial Dysbiosis in Kidney Transplant Recipients

Several transplantation-related factors can alter the gut microbiota of KTRs, such as discontinued dialysis and drugs used in ESKD, dietary changes, induction therapy, surgery, maintenance immunosuppressive therapy, and antimicrobial prophylaxis following transplantation [26]. The resulting alterations in the gut microbiome may create a signature in the microbial community that, in turn, impacts transplantation outcomes. Whether a distinctive profile can be attributed to KTRs has been gradually explored over the last decade (Table 1).

Guirong et al. showed an increased abundance of *Bacteroides* and *Enterobacteriaceae* in the gut microbiota of KTRs compared with healthy individuals. Higher *Enterobacteriaceae* abundance in KTRs was identified as a microbial signature distinctive from ESKD patients undergoing hemodialysis. *Enterobacteriaceae* is the most common cause of urinary tract infection (UTI), and higher intestinal abundance may increase the risk of post-transplant infections in KTRs. Post-transplant microbiota was also characterized by a decreased abundance of the short-chain-fatty-acid-producing bacteria *Ruminococcaceae*, *Lachnospira*, and *Faecalibacterium*. The most common SCFAs include acetate, propionate, and butyrate. SCFAs are essential in immune regulation and blood pressure control—both critical functions in KTRs [4]. Other butyrate-producing bacteria, such as *Eubacterium rectale* and *Roseburia*, have been identified to decrease in KTRs after six years post-transplantation compared with healthy individuals [5] (Table 1).

A gut microbiota analysis of KTRs at different graft periods between 3 months and 22 years post-transplantation showed decreased microbial diversity, richness, and compositional changes over time compared to healthy subjects. Proteobacteria increased shortly after transplantation and continued to increase up to 22 years post-transplantation (Table 1). At the genus level, a higher abundance of *Asteroleplasma*, *Roseburia*, *Faecalibacterium*, and *Bacteroides* characterized the first-year post-transplantation, while the gut microbiota in the long post-graft period was characterized by a higher abundance of *Rikenellaceae RC9*, *Dialister, Parabacteroides*, *Sutterella*, *Escherichia/Shigella*, and *Succinivibrio*. These results pointed out differences in the gut microbiota according to graft time, which may be a future target to predict graft function and stability over time [6].

To better evaluate whether a specific microbial signature can be identified for KTRs, few studies have compared changes in the gut microbiota pre- and post-transplantation. A pilot study analyzed the gut microbiota of five KTRs before and two weeks after KT. Although changes in richness and evenness were not detected post-transplantation, the authors observed an increase in the relative abundance of Proteobacteria and a decrease in Bacteroidetes [7] (Table 1). This imbalance could contribute to the high rate of post-transplant infections in KTRs, as Proteobacteria include numerous infectious pathogens. Moreover, decreased Bacteroidetes abundance could negatively impact immune regulation. *Bacteroides* spp. have immunomodulatory functions, such as CD4^+^ T-cell development, Th1/Th2 immune balance regulation, and IL-10 activation [27]. It has been described that *Bacteroides fragilis* mediates the conversion of CD4^+^ T cells into IL10-producing Foxp3^+^ Treg cells through the capsular polysaccharide A in a toll-like receptor-2-dependent mechanism [28]. Foxp3^+^ Treg cells and IL-10 are involved in mechanisms of allograft immune tolerance [29], and a decreased colonic abundance of *B. fragilis* could potentially contribute to allograft rejection.

Yu et al. reported a reduction in microbial richness within the first-month post-transplantation compared to the fecal microbiome one-week pre-transplantation. An alteration in the abundance of specific taxa was associated with KT, but these results should be validated in a larger population [8] (Table 1). When comparing the gut microbiota of KTRs before and up to eight weeks after transplantation, Chan et al. reported an increase in *Roseburia intestinalis* and *F. prausnitzii* following transplantation. These observations contrast with other reports suggesting a decreased abundance of these SCFA-producing bacteria [4,5] (Table 1), which may be related to the application of different sequencing and data analysis methods and the use of KTRs’ pre-transplant specimens as the control group. The study also indicated that nephrectomy had no impact on gut microbial richness or diversity because no significant changes in microbial composition were observed in kidney donors after the nephrectomy. Instead, other factors might be accounting for the alterations in KTRs, such as reduced kidney function, induction and immunosuppressive therapies, and antimicrobial prophylaxis [9].

These studies generally revealed compositional changes following transplantation, including lower richness and diversity, increased abundance of Proteobacteria, and depletion in several SCFA-producing species. However, identifying a unique profile associated with KT remains challenging due to the confounding effect of immunosuppressants, antimicrobial treatment, and dietary habits on microbiome analyses. In addition, KTRs represent a heterogeneous population diagnosed with different pathologies and comorbidities that might shape the gut microbiome differently and add complexity to post-transplant comparative analysis. Moreover, the comparison across studies is limited by the need for standard methodologies in collecting and assessing microbiome data.

It is worth noting that lower diversity and dysbiosis already characterize the gut microbiota of CKD patients before transplantation. CKD is associated with lower intestinal *Bifidobacteriaceae*, *Lactobacillaceae*, *Bacteroidaceae*, and *Prevotellaceae* and with higher intestinal levels of *Enterobacteriaceae* [30]. Studies on KTRs have reported similar results showing a decreased abundance of *Bifidobacteriaceae* and an increased abundance of *Enterobacteriaceae* [5]. This suggests that part of the dysbiosis observed in KTRs is present pre-transplantation and may not recover afterward. Moreover, as previously reported for KTRs [4,5,6], ESKD patients have a higher abundance of Proteobacteria compared with healthy individuals [31] and depletion in several SCFA-producing species, such as *Roseburia* spp., *Faecalibacterium* spp., and *E. rectale*, consistent with a reduction in SCFAs in the fecal metabolome of the patients [32]. Taking these similarities into account, it can be challenging to discern the gut microbial community of CKD and ESKD patients from that in KTRs.

## 4. The Impact of Gut Microbiota on Kidney Transplantation

Despite advances in surgical techniques, immunosuppressive therapy, and post-transplant surveillance protocols, long-term allograft survival remains challenged by several complications. KTRs can develop T-cell-mediated rejection (TCMR) and/or antibody-mediated rejection (ABMR), potentially leading to allograft failure [33]. Episodes of allograft rejection, immunosuppressive medications, and the accumulation of uremic retention solutes increase the risk of cardiovascular diseases. Moreover, immunosuppressives such as calcineurin inhibitors and corticosteroids induce metabolic complications that increase the risk of developing post-transplant diabetes. KTRs may also suffer from infectious complications, diarrhea, and cancer [34]. Hence the search for strategies that improve these patients’ quality of life, graft survival, and life expectancy. Recent studies have suggested the potential influence of gut microbiota on KT outcomes (Table 2).

### 4.1. Uremic Retention Solutes

Gut-derived uremic toxins trimethylamine N-oxide (TMAO), p-cresyl sulfate (pCS), p-cresyl glucuronide (pCG), indoxyl sulfate (IxS), and indole-3-acetic acid (IAA) have been identified to play a role in inflammation, metabolic function, cardiovascular disease, and fibrosis in CKD patients [35]. Several gut bacteria that have been found to increase after KT (Table 1) and are associated with post-transplant complications (Table 2) produce gut-derived uremic toxins precursors. Among these bacteria, *Enterobacteriaceae*, *Enterococcaceae*, *Clostridiaceae*, and *Bacteroides* sp. produce IAA, *E. coli* produces both IAA and indole, and *Clostridium tertium* and *Bacteroides* sp. produce *p*-cresol [36]. Studying these uremic toxins and their association with kidney function and graft outcome is an open area in KTRs.

TMAO plasma concentrations have been associated with an increased risk of graft failure and proposed as a potential biomarker of graft function in KTRs [37]. Other studies have reported increased serum pCS and IxS in KTRs with advanced CKD stages [38]. The treatment with a synbiotic (Probinul Neutro, CadiGroup, Rome, Italy), mainly containing *Lactobacillus* and *Bifidobacterium* spp., decreased plasma *p*-cresol in KTRs after 15 and 30 days of administration. Although the study included a small population (n = 36), the results indicate that gut microbiota interventions could reduce uremic toxicity in KTRs [39].

Liabeuf et al. reported no association between serum IxS levels and adverse outcomes post-transplantation, including graft loss, cardiovascular events, and mortality in a cohort of 311 KTRs [40]. Similarly, a study on the same cohort reported no association between serum-free and total IAA with adverse outcomes of graft loss, cardiovascular events, and death following KT [41].

### 4.2. Allograft Function

Delayed graft function is a common post-transplant event that can expose the patient to a longer uremic period and increase the risk of gut dysfunction, systemic inflammation, and allograft rejection [42]. Kim et al. showed that six-month allograft function was associated with a similar pre-transplantation microbial structure between donor and KTR, especially in genetically unrelated pairs such as spousal donor and recipient. The microbial distance in unrelated pairs was a better predictor of the eGFR than the number of human leukocytes antigen mismatches and incompatibility. This may indicate that microbial similarity between donors and KTRs could, to some extent, compensate for the disadvantages of genetic disparity and influence the criteria for living donor selection. Furthermore, a similar microbial structure between donor–KTR could potentially reduce the risk of infections, as microbial dissimilarity was associated with an increased post-transplant infection rate [43].

### 4.3. Allograft Rejection

Immune tolerance is crucial to ensure graft function and survival of transplant recipients. T-lymphocyte peripheral tolerance is the primary mode of tolerance to transplanted organs. It eliminates activated T-cell clones in the periphery via apoptosis, develops T-lymphocyte anergy, and suppresses alloreactive T lymphocytes by Tregs [44].

Members of the gut microbiota, such as *Clostridia*, *Bacteroides fragilis*, and *Bacteroides thetaiotaomicron*, have been found to modulate Tregs differentiation [45]. CD4^+^ Tregs, regulatory B cells, and dendritic cells (DCs) secrete IL-10 that inhibits APC activity and promotes the conversion of T cells into T regulatory type 1 cells (T_R_1s). T_R_1s are able to suppress the pro-inflammatory activities of both APC and effector T cells. Furthermore, in the presence of IL-10, naive CD8^+^ T cells can be converted into CD8^+^ Tregs that inhibit effector T cells. CD4^–^CD8^–^ Tregs induce effector T-cell apoptosis via the CD95-CD95L pathway and by downregulating the expression of DCs molecules CD80 and CD86, consequently inhibiting the ability of DCs to stimulate pro-inflammatory responses. Tolerogenic DCs inhibit effector T-cell proliferation and differentiation into Th1 and Th17 cells [29].

The gut microbiota and derived metabolites can also modulate host inflammatory response through microbiota–cytokine interactions [46]. Schirmer et al. observed that interindividual variation in cytokine response to various microbial stimulations was linked to specific commensal bacteria and microbial functions in healthy individuals. For instance, a higher abundance of *Roseburia* was associated with lower IL-6 levels, and a higher abundance of *Bilophila* and *Odoribacter* was associated with lower TNFα levels. Furthermore, lower IL-17 production was linked to a higher abundance of *Faecalibacterium* and *Atopobium*, whereas *Escherichia*, *Anaerotruncus*, *Coprobacillus*, and *Clostridium* higher abundance was linked to increased production of IL-17. These results suggest that modulating cytokine expression by microbial interventions may have therapeutic value in KTRs. However, microbiota–cytokine interactions should be evaluated in KT, given the stimulus-specific and/or cytokine-specific nature of these interactions [47].

To date, few studies have addressed the potential association between gut microbiota and allograft rejection in KTRs. Wang et al. analyzed the gut microbial community of KTRs with ABMR compared to KTRs who did not develop graft rejection. ABMR was related to lower microbial richness and decreased relative abundance of *Clostridia*, *Paraprevotellaceae*, and *Faecalibacterium*, as well as increased abundance of *Enterococcaceae*, *Coprobacillus*, and *Enterobacter*, among other taxa (Table 2). These microbial changes are likely associated with ABMR because recipients with a recent history of infection, antibiotic usage, gastric/colon resection, and non-infectious diarrhea were excluded from the study. However, immunosuppressive therapy could have contributed to the alterations observed. *Clostridiales* was proposed as a potential biomarker of ABMR after KT, but further studies are needed to validate its use as a diagnostic tool [10].

Lee et al. also reported alterations in the gut microbiota of KTRs associated with acute rejection (AR). During the first three months post-transplantation, three patients of the studied cohort were diagnosed with ABMR, TCMR, and mixed ABMR-TCMR, respectively. A lower abundance of Clostridiales, Bacteroidales, *Eubacterium dolichum*, and *Ruminococcus,* and a higher abundance of Lactobacillales, *Enterococcus*, *Anaerofilum*, and *Clostridium tertium* characterized the gut microbiota of AR patients compared to those who did not develop AR (Table 2). It should be noted that AR patients received antibiotic therapy to treat *Clostridioides difficile* infection (CDI), Enterococcus UTI, *Escherichia coli* UTI, and *Klebsiella*/*Serratia* UTI before AR. It is then difficult to determine whether the observed microbial alterations are related to AR or the result of antibiotic therapy [7].

Another study analyzed the pre-and post-transplant rectal microbiota of four KTRs who experienced rejection compared to KTRs who did not experience rejection or other adverse events. A decreased abundance of *Anaerotruncus*, *Coprobacillus*, *Coprococcus*, and an unknown member of *Peptostreptococcaceae* correlated with the development of future rejection events [13] (Table 2). However, these results require further validation in a larger cohort to assess the potential of gut microbial taxa as biomarkers of rejection events.

### 4.4. Immunosuppressants Metabolism

Current clinical guidelines recommend combining immunosuppressive medications as maintenance therapy, including a calcineurin inhibitor and an antiproliferative agent with or without corticosteroids [3]. Recent data suggest the impact of the gut microbiota on the most prescribed immunosuppressive medications: MMF and TAC.

The use of MMF has been correlated to a lower diversity of the gut microbiome [5], post-transplant diarrhea, and impaired quality of life in KTRs [48]. The gut microbiome can also metabolize MMF, thereby influencing drug dosage. MMF is converted to its active form, mycophenolic acid (MPA), by plasma and tissue esterases and inactivated by hepatic glucuronidation to MPA glucuronide. MPA glucuronide is then excreted in the urine and bile. If secreted into the gastrointestinal tract through the ATP-binding cassette subfamily C member 2 protein, bacteria expressing beta-glucuronidase enzymes can cleave the glucuronic acid (GA) of MPA glucuronide to produce free MPA and GA. The resulting GA is a carbon source for bacterial metabolism, and MPA undergoes enterohepatic recirculation, which has been related to gastrointestinal toxicity [21,26]. Hence, the management strategies for gastrointestinal complications include MMF dose reduction or discontinuation, which can lead to an increased risk of allograft rejection [49].

TAC is a macrolide of the calcineurin inhibitor family that binds to the FK506-binding protein, forming a complex that inhibits calcineurin phosphatase, ultimately blocking T-cell activation. Despite its efficacy in avoiding TCMR and ABMR, TAC decreases eGFR and promotes glucose intolerance, new-onset diabetes, and hypertension [2,3]. TAC has also been associated with gastrointestinal symptoms in KTRs, including diarrhea, nausea, and vomiting [48]. Its antimicrobial activity can disrupt the gut microbial community, and recent data indicate a bidirectional relationship as gut bacteria metabolize TAC.

Lee et al. observed that the fecal abundance of *F. prausnitzii* was positively correlated with the one-month TAC dose required to maintain therapeutic levels in KTRs. This suggested a possible role of *F. prausnitzii* influencing TAC levels, but the underlying mechanisms were not identified [2]. A follow-up study found that *F. prausnitzii* and other commensal bacteria, mainly belonging to Clostridiales, metabolize TAC into a less effective immunosuppressive metabolite (M1). M1 is a C9 keto-reduction product, uniquely synthesized by gut bacteria and 15-fold less immunosuppressive than its parent in inhibiting T-lymphocyte proliferation in vitro. However, the authors observed no correlation between Clostridiales abundance (including *F. prausnitzii*) and M1 production in stool samples from KTRs undergoing oral TAC therapy [11]. In a subsequent study, the group showed active metabolism of TAC in KTRs by evaluating the pharmacokinetics of M1 after oral administration of TAC. M1 was detected within the first four hours of administration, with concentrations reduced by at least five-fold compared to parent TAC. These results could explain the interpatient variability in TAC therapeutic level requirements and suggest that changes in the gut microbiota might impact TAC trough variability [12] (Table 2).

### 4.5. Post-Transplant Infection

Urinary tract infections remain among the most frequent complications affecting post-transplant patients [50]. A pilot study reported an increased fecal abundance of *Enterococcus* correlated with the development of *Enterococcus* UTI in KTRs (Table 2). However, this observation was limited by the confounding effect of immunosuppressives, antimicrobial therapy, and the small sample size studied: 3 KTRs *Enterococcus* UTI compared with 23 KTRs who did not develop *Enterococcus* UTI [7].

Fricke et al. described alterations in the rectal microbiota of KTRs associated with urinary and upper respiratory tract infections during the first six months post-transplantation (Table 2). A potential role in predicting post-transplant infection events was attributed to *Anaerotruncus* as it significantly decreased in 4 KTRs with infection compared with 14 KTRs without post-transplant adverse events. Nonetheless, the value of these microbial alterations as diagnosis markers requires further evaluation in a larger population [13].

Another study serially profiled fecal specimens from KTRs within the first three months after transplantation (Table 2). The authors observed 1% *Escherichia* gut abundance associated with the future development of *Escherichia* bacteriuria. Female gender was also a predictor of future *Escherichia* bacteriuria. Likewise, 1% *Enterococcus* gut abundance was associated with the future development of *Enterococcus* bacteriuria, independent from other factors such as gender, antimicrobial treatment, and immune maintenance therapy. The phylogenetic analysis showed a close relationship between the same subject’s urine and fecal strains of *E. coli*, *Enterococcus faecalis*, and *Enterococcus faecium*. The similarity of *E. coli* strains was supported by detecting uropathogenic genes and beta-lactams, sulfonamides, and trimethoprim resistance genes in paired urine and fecal specimens associated with *E. coli* bacteriuria. These results suggest that an overgrowth of enteropathogenic bacteria could influence UTI development in KTRs [14].

In a follow-up study, the group found that a high abundance of *Faecalibacterium* and *Romboutsia* was significantly associated with a lower risk of developing *Enterobacteriaceae* bacteriuria and *Enterobacteriaceae* UTI. On the other hand, increased Lactobacillus abundance was associated with an increased risk of developing *Enterobacteriaceae* bacteriuria and *Enterobacteriaceae* UTI [15] (Table 2). These results are promising for developing personalized UTI treatments such as *Enterobacteriaceae*-reducing probiotics.

Analyzing the previous cohort, the authors suggested a possible protective role of butyrate-producing bacteria in developing respiratory viral infections. A fecal abundance of butyrate-producing bacteria higher than 1% was associated with a decreased risk of developing respiratory viral infections in the first two years post-transplantation and CMV viremia in the first-year post-transplantation [16] (Table 2). Butyrate has important roles in immunomodulation, maintenance of the intestinal barrier, and protection against bacterial and viral infections. Another study in allogeneic hematopoietic stem cell patients indicated that a high intestinal abundance of butyrate-producing bacteria at the time of engraftment conferred protection against viral lower respiratory tract infections [51]. However, these studies could not determine a causal relationship between butyrate-producing bacteria and protection against respiratory viral infections.

### 4.6. Post-Transplant Diarrhea

Post-transplant diarrhea increases the risk of graft failure, death-censored graft survival, and patient death. Despite being a common complication, the etiology of post-transplant diarrhea is not identified in most cases, and it is frequently associated with intestinal drug toxicity if infections are excluded. A retrospective study reported that over 80% of KTRs were diagnosed with unspecified non-infectious diarrhea, with greater incidence in patients following TAC and MMF combined therapy [52]. Recent data suggest alterations in the gut microbiome of KTRs as a potential non-infectious etiology of post-transplant diarrhea.

In a pilot study, Lee et al. observed a lower gut microbial diversity and a decrease in the relative abundance of *Bacteroides*, *Ruminococcus*, *Coprococcus*, and *Dorea* to be associated with the development of diarrhea within the first-month post-transplantation [7]. Analyzing a different cohort, the group found that the development of post-transplant diarrhea was associated with gut dysbiosis instead of infectious etiologies. The gut microbiota of KTRs with diarrhea was characterized by a decreased abundance of *Ruminococcus*, *Coprococcus*, *Dorea, Faecalibacterium*, and *Bifidobacterium*, as well as an increased abundance of *Escherichia* and *Enterococcus*. The potential role of these taxa on post-transplant diarrhea was also suggested by the gut microbial analysis of two KTRs with a history of CDI and who underwent fecal microbial transplantation (FMT). The patients had persistent diarrhea despite testing negative for *C. difficile* before FMT. After FMT, the resolution of diarrhea correlated with an overall increase in the abundance of the taxa previously identified as significantly lower in diarrheal specimens and an overall decrease in the abundance of the taxa identified as significantly higher in diarrheal specimens [17] (Table 2). These results indicate that restoring the gut microbial community imbalance could successfully manage post-transplant diarrhea.

A further study by the group reported a lower abundance of *Ruminococcus*, *Anaerostipes*, *Fusicatenibacter*, *Eubacterium*, *Ruminiclostridium*, *Dorea*, and *Bifidobacterium* associated with non-infectious diarrhea episodes in KTRs. Moreover, prolonged diarrhea was associated with higher beta-glucuronidase activity, indicating that the toxicity from the free MPA in the colon could contribute to the diarrhea episodes, though these results require further validation. Four genera were positively correlated with beta-glucuronidase activity: *Subdoligranulum*, *Coprococcus*, *Tyzzerella*, and an unspecified *Erysipelotrichaceae* [18] (Table 2). These studies denote a potential relationship between the gut microbiota and the development of post-transplant diarrhea up to three months after KT.

### 4.7. New Onset Diabetes (NODAT)

NODAT develops in approximately 20% of KTRs in the first year after transplantation and has been identified as an adverse effect of immunosuppressive treatment, including corticosteroids, cyclosporin, TAC, and sirolimus [2,19]. Lecronier et al. observed alterations in the gut microbiota associated with the development of NODAT after KT by comparing pre- and post-transplant fecal samples from KTRs. An increase in *Lactobacillus* sp. relative abundance and a decrease in *Akkermansia muciniphila* were associated with NODAT presentation (Table 2). The same microbial changes were observed in patients with pre-transplant diabetes but not in patients without diabetes either before or after KT, suggesting a potential role for these taxa in the future development of NODAT. It should be noted that the results were obtained by qPCR of targeted bacterial species. Other taxonomic changes may be revealed through metagenomic analyses. Moreover, other factors could contribute to NODAT presentation, such as immunosuppressive medications and increased body mass index of the patients following KT, because obesity is a known risk factor for diabetes. Additional studies should validate the possible role of gut microbiota in NODAT development in KTRs [19].

## 5. Gut-Microbiota-Based Therapies in Kidney Transplantation

The plasticity of the gut microbiome allows the development of therapeutic interventions to prevent and treat health disorders. Diet intervention and FMT are approaches to reshape the entire gut microbiome, while prebiotics, probiotics, and bacteriophages are more targeted manipulations [53]. FMT and probiotics have been used to decrease the risk of recurrent CDI. A year of prophylaxis with the probiotic *Lactobacillus plantarum* 299v (LP299v) decreased CDI incidence in immunosuppressed patients receiving antibiotics therapy. Furthermore, LP299v prophylaxis was associated with reduced diarrhea events and lower serum C-reactive protein concentrations. A year of follow-up upon stopping LP299v use showed increased CDI incidence, suggesting that this probiotic may prevent CDI in immunosuppressed patients [54].

The administration of probiotics has also shown immunomodulatory effects. A mixture containing *Lactobacillus acidophilus*, *Lactobacillus casei*, *Lactobacillus reuteri*, *Bifidobacterium bifidium*, and *Streptococcus thermophilus* was studied in an inflammatory bowel disease mice model. The probiotics induced regulatory DCs that promoted the generation and migration of CD4^+^Foxp3^+^ Tregs to the inflammation sites. In addition, the probiotics suppressed the expression of inflammatory cytokines IL-17, IFNγ, and TNFα in T and B cells and enhanced the suppression capacity of naturally occurring CD4^+^CD25^+^ Tregs [55]. The generation of CD4^+^Foxp3^+^ Tregs in response to probiotics may be therapeutically beneficial in stimulating kidney allograft tolerance.

The application of gut microbiota-based therapies has not been widely explored in KT. Recent data, mainly in case reports, suggest the efficacy of these therapies for treating and preventing infectious complications. A case reported a heart and kidney transplant recipient who underwent allogeneic FMT because of recurrent episodes of *C. difficile* colitis, vancomycin-resistant Enterococcus UTIs, and bacteremia. FMT resulted in a decreased *Enterococcus* abundance and the resolution of *C. difficile* and vancomycin-resistant *Enterococcus* infections up to one year of follow-up [22]. Another study reported a KTR with a history of UTIs during the three first years of graft function, predominantly *E. coli* and ESBL-producing *E. coli*. A gradual decrease in *Enterobacteriaceae* relative abundance was observed following FMT. The patient had no recurrent *E. coli* UTI during the nine-month follow-up [23].

Successful FMT was also reported for a KTR with recurrent UTI caused by ESBL-producing *Klebsiella pneumoniae* (ESBL-*K. pneumoniae*), CDI, and who was suffering from diarrhea. Although six days after FMT, the patient was re-admitted with ESBL-*K. pneumoniae* infection, and no further episodes of UTI occurred after his recovery. A resolution of diarrhea was achieved, and no UTIs or CDI symptoms were detected during the first year post-FMT [24].

Wang et al. conducted a successful FMT on a KTR with carbapenem-resistant *K. pneumoniae* infection detected in incision secretion, urine, and rectal swab cultures. The strains causing the infection were clustered phylogenetically closed, suggesting the potential involvement of gut bacteria in post-transplant infections. One week after FMT, the urine and rectal swab cultures tested negative for *K. pneumoniae* infection, and the surgical incision healed after seventeen days. The patient had no symptoms of infection during the two months of follow-up [25].

Despite the promise of FMT to treat post-transplant infections and diarrhea in KTRs, this therapy should be used cautiously. A study reported two cases of ESBL-producing *E. coli* bacteremia transmitted through FMT. One of the cases resulted in death, stressing how important donor selection is to avoid adverse events [56]. The future application of FMT may involve the development of defined microbial mixtures to prevent unfavorable clinical outcomes.

## 6. Conclusions and Future Perspectives

The gut microbiota is increasingly recognized as influencing kidney post-transplant outcomes. Although no unique microbial signature can be attributed to KTRs thus far, KT results in lower gut microbial richness and diversity than the healthy population. Specific taxa have been associated with the development of infections and diarrhea complications, allograft rejection, and the ability to metabolize immunosuppressants essential for preserving graft function. These findings are promising to personalize immunosuppressive therapies based on interpersonal microbiome variability. In addition, microbiota intervention therapies such as FMT have successfully resolved infection and diarrhea complications in KTRs. These data suggest the potential role of gut microbiota in graft survival. However, we have just begun to understand the role of the gut microbiome in KT. More studies are needed to define the community structures representative of a healthy microbiome in transplant patients early after transplantation and longitudinally, as well as the role of gut-microbiome-modulating alloimmunity.

## Figures and Tables

**Figure 1 biology-12-00163-f001:**
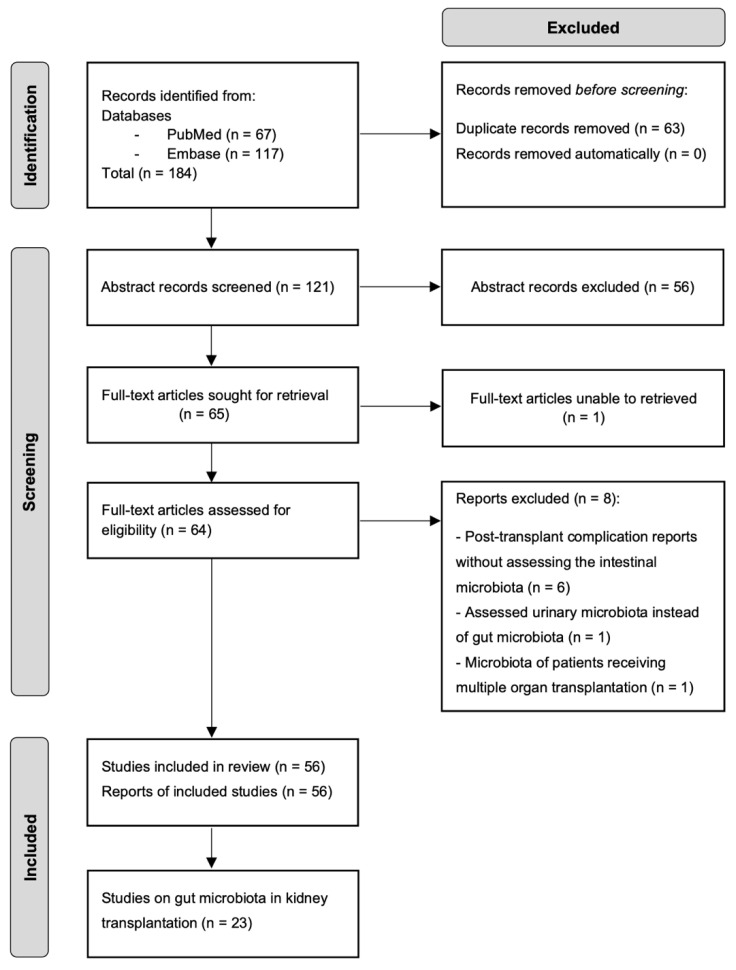
Flowchart of the literature search on the role of gut microbiota in kidney transplantation.

**Table 1 biology-12-00163-t001:** Gut Microbiota Composition in Kidney Transplant Recipients.

Study Population	Sample	Method of Detection	Gut Microbiota Abundance after KT		Reference
KTRs (N = 16) HD (N = 84) HC (N = 53)	Feces	V3 16SrRNA sequencing; Ion Personal Genome Machine	Family: ↑ *Enterobacteriaceae* Genus: ↑ *Bacteroides*	↓ *Ruminococcaceae* ↓ *Lachnospira* ↓ *Faecalibacterium*	[4]
KTRs (N = 139) HC (N = 105)	Feces	V4-V5 16SrRNA sequencing; Illumina MiSeq	Phylum: ↑ Proteobacteria Species: ↑ *Escherichia coli* ↑ *Streptococcus thermophilus* ↑ *Streptococcus mitis* ↑ *Streptococcus parasanguinis* ↑ *Blautia faecis* ↑ *Blautia glucerasea*	↓ Actinobacteria ↓ *Bifidobacterium* spp. ↓ *Ruminococcus bromii* ↓ *Faecalibacterium prausnitzii* ↓ *Coprococcus eutactus* ↓ *Eubacterium siraeum* ↓ *Eubacterium rectale* ↓ *Dorea longicatena* ↓ *Coprococcus catus* ↓ *Coprococcus comes* ↓ *Roseburia* sp.	[5]
KTRs (N = 40) HC (N = 18)	Feces	V3-V4 16SrRNA sequencing; Illumina MiSeq	Phylum: ↑ Proteobacteria Genus: ↑ *Bacteroides* ↑ *Faecalibacterium* ↑ *Escherichia/Shigella* ↑ *Roseburia* ↑ *Succinivibrio*	↓ Actinobacteria ↓ Verrucomicrobia ↓ *Ruminococcaceae UCG.002* ↓ *Clostridium sensu strico 1* ↓ *Subdoligranulum* ↓ *Dialister* ↓ *Parabacteroides* ↓ *Alistipes* ↓ *Prevotella 9*	[6]
KTRs (N = 26)	Feces	V4-V5 16SrRNA sequencing; Illumina MiSeq	Phylum: ↑ Proteobacteria Order: ↑ Erysipelotrichales ↑ Enterobacteriales	↓ Bacteroidetes	[7]
KTRs (N = 10)	Feces	V4-V5 16SrRNA sequencing; Illumina MiSeq	Class: ↑ Bacilli Order: Family: ↑ *Enterococcaceae* Genus: ↑ *Enterococcus* ↑*Anaerostipes*	↓ Clostridiales ↓ *Ruminococcaceae* ↓ *Veillonellaceae* ↓ *Faecalibacterium*	[8]
KTRs (N = 15) Donors (N = 15)	Feces	Metagenomic sequencing; Illumina NovaSeq 6000	Family: Genus: ↑ *Roseburia* ↑ *Streptococcus* ↑ *Oscillibacter* ↑ *Romboutsia* ↑ *Pauljensenia* Species: ↑ *Roseburia intestinalis* ↑ *Faecalibacterium prausnitzii*	↓ *Acutalibacteracae* ↓ *Rikenellacea*	[9]

KT—kidney transplant; KTRs—kidney transplant recipients; HD—hemodialysis; HC—healthy controls; ↑ Increased relative abundance; ↓ Decreased relative abundance.

**Table 2 biology-12-00163-t002:** The Role of Gut Microbiota in Kidney Transplantation.

Post-Transplant Setting	Study Population	Graft Time	Gut Bacteria Involved		Outcome	Reference
TAC Dosing	KTRs (N = 19)	1 month	↑ *Faecalibacterium prausnitzii*		Increased abundance positively correlated with increased TAC dose requirements.	[3]
Infection	KTRs (N = 26) - Infection (N = 3)	3 months	↑ *Enterococcus*		Increased abundance associated with the development of Enterococcus UTI.	[7]
Diarrhea	- Diarrhea (N = 6) - No Diarrhea (N = 9)			*↓ Bacteroides* *↓ Ruminococcus* *↓ Coprococcus* *↓ Dorea*	Decreased abundance associated with the development of post-transplant diarrhea.	
Rejection	Rejection (N = 3)		↑ *Lactobacillales* ↑ *Enterococcus* ↑ *Anaerofilum* ↑ *Clostridium tertium*	*↓* Clostridiales *↓* Bacteroidales *↓ Lachnospiraceae* *↓ Blautia* *↓ Eubacterium dolichum* *↓ Ruminococcus*	Changes in the relative abundance associated with the development of acute rejection.	
Rejection	KTRs (N = 53) - KTRs ABMR (N = 24) - KTRs No ABMR (N = 29)		↑ *Coprobacillus* ↑ *Serratia* ↑ *Thermus* ↑ *Atopobium* ↑ *Enterococcus* ↑ *Rothia* ↑ *Granulicatella* ↑ *Enterobacter* ↑ *Eubacterium* ↑ *Epulopiscium*	*↓* Clostridiales *↓ Barnesiellaceae* *↓ Paraprevotellaceae* *↓ Pasteurellaceae* *↓ Roseburia* *↓ Haemophilus* *↓ Faecalibacterium* *↓ Paraprevotella*	Gut microbiota alterations associated with ABMR.	[10]
TAC Metabolism	In vitro (microbial culture)		*- Faecalibacterium prausnitzii* (Clostridiales) - Erysipelotrichales - Bacteroidales		Taxa able to metabolize TAC into a less effective immunosuppressant metabolite (M1).	[11]
TAC Metabolism	KTRs (N = 10)		Gut bacteria		Active metabolism of TAC by the gut bacteria. The gut microbiota could impact TAC trough variability.	[12]
Infection	KTRs (N = 60) - Infection (N = 4) - No infection (N = 14)	6 months		*↓* Clostridiales *↓ Mogibacterium* *↓ Peptoniphilus* *↓ Coriobacterineae*	Changes in the relative abundance associated with the development of infections after six months post transplantation.	[13]
Rejection	- Rejection (N = 4) - No rejection (N = 14)			*↓ Anaerotruncus* *↓ Coprobacillus* *↓ Coprococcus* *↓ Peptostreptococcaceae* sp.	Decreased relative abundance correlated with future development of rejection events.	
Infection	KTRs (N = 168)*	3 months	↑ *Escherichia* ↑ *Enterococcus*		Increased abundance associated with the development of *Escherichia* and *Enterococcus* bacteriuria.	[14]
Infection	KTRs (N = 168)*	3 months	↑ *Faecalibacterium* ↑ *Romboutsia*		Increased abundance associated with lower risk of Enterobacteriaceae bacteriuria and UTI.	[15]
			↑ *Lactobacillus*		Increased abundance associated with higher risk of Enterobacteriaceae bacteriuria and UTI.	
Infection	KTRs (N = 168)*	3 months	Butyrate-producing bacteria		A relative abundance higher than 1% associated with lower risk of respiratory viral infection and CMV viremia.	[16]
Diarrhea	KTRs (N = 64) - Diarrhea (N = 18) - No Diarrhea (N = 46)	3 months	↑ *Enterococcus* ↑ *Escherichia* ↑ *Lachnoclostridium*	*↓ Eubacterium* *↓ Anaerostipes* *↓ Coprococcus* *↓ Romboutsia* *↓ Ruminococcus* *↓ Dorea* *↓ Faecalibacterium* *↓ Fusicatenibacter* *↓ Oscillibacter* *↓ Ruminiclostridium* *↓ Blautia* *↓ Bifidobacterium* *↓ Bacteroides*	Changes in the relative abundance associated with the development of diarrhea.	[17]
Diarrhea	KTRs (N = 79) - Diarrhea (N = 22) - No Diarrhea (N = 57)	3 months		*↓ Eubacterium* *↓ Anaerostipes* *↓ Ruminococcus* *↓ Dorea* *↓ Fusicatenibacter* *V Ruminiclostridium* *↓ Bifidobacterium*	Decreased relative abundance associated with the development of non-infectious diarrhea.	[18]
			- *Subdoligranulum* - *Coprococcus* - *Tyzzerella* - *Erysipelotrichaceae* sp.		Relative abundance associated with β-glucuronidase activity, which in turn is associated with prolonged diarrhea.	
NODAT	KTRs (N = 50) - NODAT (N = 15) - Initial Diabetes (N = 16) - No Diabetes (N = 19)	9 months	↑ *Lactobacillus* sp.	*↓* *Akkermansia muciniphila*	Changes in the relative abundance associated with the development of NODAT.	[19]

KTRs—kidney transplant recipients; TAC—Tacrolimus; UTI—urinary tract infection; CMV—cytomegalovirus; ABMR—antibody-mediated rejection; and NODAT—new-onset diabetes. * Same cohort was studied. ↑ Increased relative abundance; ↓ Decreased relative abundance.

## Data Availability

Not applicable.

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
