# Peer review of "The Gut Microbiota in Kidney Transplantation: A Target for Personalized Therapy?"

_biology, 2023, doi:10.3390/biology12020163_

Round 1

Reviewer 1 Report

This review presents recent knowledges concerning gut microbiota, immune system cells and immunosuppressive therapy in patients underlying kidney transplantation. The work is well presented, and the manuscript is well organized. However, some remarks and comments need to be answered as followed.

1-      Recent studies have shown that gut microbiota play an important role the modification of host inflammatory cytokines. This effect could affect kidney transplantation. The authors should describe this effect, at least,  in a small section.

2-      The section 4 seems to be not detailed in comparison to sections 2 and 3. The authors should present in section 4, if possible, more potential therapeutic targets to personalize immunosuppressive therapy.

3-      A preview review published recently (Transplantation Reviews 2022) is entitled “Impact of gut microbiota on kidney transplantation”. The authors should modify the title of the manuscript to avoid any confusion.

4-      Several abbreviations are indicated in the manuscript. The authors should limit them for those mostly used and known. For example, the authors should not need to define TAC for tacrolimus.

Author Response

Point 1: Recent studies have shown that gut microbiota plays an important role the modification of host inflammatory cytokines. This effect could affect kidney transplantation. The authors should describe this effect, at least, in a small section.

Response 1: We agree with this comment and thank the Reviewer for the suggestion. Several studies have reported a potential role of the gut microbiome in modulating inflammatory pathways such as cytokine production. These studies have been evaluated in experimental mice models of ulcerative colitis and goat, inflammatory bowel diseases, and colorectal cancer [46]. In healthy individuals, specific microbial communities and functions were associated with variations in cytokine response among individuals, suggesting the cytokine regulation capacity of the gut microbiome. We decided to acknowledge this data in our review, given that the pro-inflammatory/regulatory cytokine network contributes to allograft rejection or tolerance in kidney transplant patients [47]. However, the studies suggest that microbiota-cytokine interactions are stimulus-specific and/or cytokine-specific. Therefore, we recommend evaluating this association in kidney transplantation to be able to consider its therapeutic value. Therefore, we added a new paragraph (third paragraph, subsection 4.3. “Allograft rejection”) with the following considerations.

The gut microbiota and derived metabolites can also modulate host inflammatory response through microbiota-cytokine interactions [46]. Schirmer et al. observed that inter-individual variation in cytokine response to various microbial stimulations was linked to specific commensal bacteria and microbial functions in healthy individuals. For instance, a higher abundance of Roseburia was associated with lower IL-6 levels, and a higher abundance of Bilophila and Odoribacter was associated with lower TNFα levels. Furthermore, lower IL-17 production was linked to a higher abundance of Faecalibacterium and Atopobium, whereas Escherichia, Anaerotruncus, Coprobacillus, and Clostridium higher abundance was linked to increased production of IL-17 [47]. These results suggest that modulating cytokine expression by microbial interventions may have therapeutic value in KTRs. However, microbiota-cytokine interactions should be evaluated in KT, given the stimulus-specific and/or cytokine-specific nature of these interactions.

Point 2: The section 4 seems to be not detailed in comparison to sections 2 and 3. The authors should present in section 4, if possible, more potential therapeutic targets to personalize immunosuppressive therapy.

Response 2: Section 4 (section 5 in the updated version) is indeed less detailed. Studies reporting the application of gut microbiota–based therapies are scarce in kidney transplantation. Most are restricted to applying fecal microbiota transplantation in recurrent infections and refractory diarrhea [22-25]. We included the use of probiotics to treat Clostridioides difficile infection in immunosuppressed patients, considering that it may also have a beneficial effect in kidney transplant patients [54]. In addition, following the Reviewer’s suggestion to improve the section, we included a report of probiotics that have shown immunomodulatory effects [55]. This might be worth exploring in kidney transplantation (second paragraph, section 5. “Gut microbiota intervention therapies in kidney transplantation”). The following description was added.

The administration of probiotics has also shown immunomodulatory effects. A mixture containing Lactobacillus acidophilus, Lactobacillus casei, Lactobacillus reuteri, Bifidobacterium bifidium, and Streptococcus thermophilus was studied in an Inflammatory Bowel Disease mice model. The probiotics induced regulatory DCs that promoted the generation and migration of CD4+Foxp3+ Tregs to the inflammation sites. In addition, the probiotics suppressed the expression of inflammatory cytokines IL-17, IFNγ, and TNFα in T and B cells, as well as enhanced the suppression capacity of naturally occurring CD4+CD25+ Tregs [55]. The generation of CD4+Foxp3+ Tregs in response to probiotics may be therapeutically beneficial in stimulating kidney allograft tolerance.

Point 3: A preview review published recently (Transplantation Reviews 2022) is entitled “Impact of gut microbiota on kidney transplantation”. The authors should modify the title of the manuscript to avoid any confusion.

Response 3: We have replaced the title with “The gut microbiota in kidney transplantation: a target for personalized therapy?”

Point 4: Several abbreviations are indicated in the manuscript. The authors should limit them for those mostly used and known. For example, the authors should not need to define TAC for tacrolimus.

Response 4: Thank you for this suggestion. The most commonly used terms were removed, and we kept the following abbreviations: KT, ESKD, KTRs, Tregs, CKD, UTI, APC, DCs, GA, FMT, NODAT, and LP299v.

References

  1. Stripling et al. Open Forum Infect Dis 2015, 2, doi:10.1093/OFID/OFV078.
  2. Biehl et al. Infection 2018, 46, 871–874, doi:10.1007/S15010-018-1190-9.
  3. Grosen et al. Case Rep Nephrol Dial 2019, 9, 102–107, doi:10.1159/000502336.
  4. Wang et al. Infect Drug Resist 2021, 14, 1805–1811, doi:10.2147/IDR.S308308.
  5. Mendes et al. J Interferon Cytokine Res 2019, 39, 393–409, doi:10.1089/JIR.2019.0011.
  6. Schirmer et al. Cell 2016, 167, 1125, doi:10.1016/J.CELL.2016.10.020.
  7. Dudzicz et al. Nutrients 2018, 10, doi:10.3390/NU10111574.
  8. Kwon et al. Proc Natl Acad Sci U S A 2010, 107, 2159, doi:10.1073/PNAS.0904055107.

Reviewer 2 Report

This is an interesting topic and the authors reviewed the role of gut microbiota in kidney transplantation. 

comments:

In the "4. Gut microbiota intervention therapies", the authors are suggested to review the effect of immunosuppressive medications on gut microbiota.

Figure 1 showed the "Mechanisms of allograft immune regulation by Treg cells", and this figure is not closely related to this topic. The authors are suggested to delete it or draw a new figure that shows the gut microbiota.

Author Response

Point 1: In the "4. Gut microbiota intervention therapies", the authors are suggested to review the effect of immunosuppressive medications on gut microbiota.

Response 1: We thank this Reviewer for the comments and suggestions. We modified the section 5 title (formerly section 4) to “Gut microbiota-based therapies in kidney transplantation,” which may better indicate the scope of the section. We aimed to describe therapies modulating the gut microbiota applied in kidney transplant patients. Although immunosuppressive drugs shape the microbial community, we did not intend to elaborate on this subject in section 5. Specific alterations in the microbial community associated with immunosuppressive therapy were explored throughout the manuscript as we described the gut bacteria related to kidney transplantation. The gut microbiota profile cannot be isolated from the confounding effects of immunosuppressive drugs and antimicrobials, among other factors. Therefore, the microbial communities described can be influenced, to some extent, by immunosuppressive therapy. This was also the case when analyzing the microbial structure associated with post-transplant complications, including allograft rejection, post-transplant diarrhea, and new-onset diabetes.

Point 2: Figure 1 showed the "Mechanisms of allograft immune regulation by Treg cells", and this figure is not closely related to this topic. The authors are suggested to delete it or draw a new figure that shows the gut microbiota.

Response 2: We replaced Figure 1 with the flow diagram of the literature search conducted.

Reviewer 3 Report

The gut microbiota is increasingly recognized as influencing kidney post-transplant outcomes. In this manuscript, the role of gut microbiota in kidney transplantation was studied well. But there are some questions in the aspects of introduction, results and discussion and so on.

Hence, I have some suggestions as follows:

1) Some descriptions in the manuscript were not exact or confusing. Some words which will make the manuscript feel like an article on a popular science book should not appear in such a research paper. The following are suggestions for improving English usage. Please use standard expression in English. 

2) The title is not good, which is too general.

3) The manuscript stays within a stage of literature survey, and is hard to find original contribution of the authors on this subject.

4) Problems on format or details: the manuscript was not well prepared according to the “Guidelines”. Please check carefully.

5) The References are too littler to support the general title.

6) If you put some photos of your manuscript into the paper, the design of your review will be more clearly understood.

Author Response

Point 1: Some descriptions in the manuscript were not exact or confusing. Some words which will make the manuscript feel like an article on a popular science book should not appear in such a research paper. The following are suggestions for improving English usage. Please use standard expression in English. 

Response 1: We thank this Reviewer for the comments and suggestions. A native English speaker proofread the manuscript to improve English usage.

Point 2: The title is not good, which is too general.

Response 2: We have replaced the title with “The gut microbiota in kidney transplantation: a target for personalized therapy?”

Point 3: The manuscript stays within a stage of literature survey, and is hard to find original contribution of the authors on this subject.

Response 3: The impact of the gut microbiome in kidney transplantation is a novel research area explored over the last decade. The scope of our review was to shed light on the open issue of using microbiota-based therapies to personalize immunosuppressive therapy and post-transplant complications based on inter-patient microbial variability. To our knowledge, this subject has not been thoroughly addressed in previous reports. Therefore, we analyzed the current data from the perspective of personalizing therapy in various contexts, including uremic toxicity, allograft immunoregulation, immunosuppressive treatment, metabolic alterations, post-transplant infections, and gastrointestinal toxicity. Other reposts have assessed these post-transplant settings individually and without the potential of developing personalized treatments in perspective.  

Point 4: Problems on format or details: the manuscript was not well prepared according to the “Guidelines”. Please check carefully.

Response 4: We have updated the Biology template, added the “Simple Summary” section, and corrected the manuscript length. In addition, we modified the sections' order by placing “Review strategy” right after the introduction instead of at the end of the manuscript.

 Point 5: The References are too littler to support the general title.

Response 5: The number of references may support a more specific title such as “The gut microbiota in kidney transplantation: a target for personalized therapy?”

Point 6: If you put some photos of your manuscript into the paper, the design of your review will be more clearly understood.

Response 6: To improve the overall clarity and meet the design requirements, we moved the Literature search flowchart, formerly as supplemental material, into the main text in section 2. “Review strategy”.

Round 2

Reviewer 3 Report

Can be published.